Familiarity breeds content: assessing bird species popularity with culturomics

http://orcid.org/0000-0001-7359-9091 Correia Ricardo A. 1 2 rahc85@gmail.com
Jepson Paul R. 2
Malhado Ana C. M. 1
Ladle Richard J. 1 2
1 Institute of Biological and Health Sciences, Federal University of Alagoas , Maceió, Alagoas , Brazil
2 School of Geography and the Environment, University of Oxford , Oxford , United Kingdom
Costanza Robert
Electronic publication date: 2016 Feb 25
Publication date: 2016
Volume: 4
Electronic Location ID: e1728
Received 2015 Dec 4; Accepted 2016 Feb 2
Copyright: © 2016 Correia et al.
Copyright year: 2016
Copyright holder: Correia et al.
License: This is an open access article distributed under the terms of the Creative Commons Attribution License, which permits unrestricted use, distribution, reproduction and adaptation in any medium and for any purpose provided that it is properly attributed. For attribution, the original author(s), title, publication source (PeerJ) and either DOI or URL of the article must be cited.
License URL: https://creativecommons.org/licenses/by/4.0/

Keywords: Birds, Public perception, Biodiversity, Culturalness, Internet salience, Culturomics, Conservation

Funding: Brazilian National Council PDJ #163055/2014-9, PVE #400325/2014-4 and #448966/2014-0 Funding was provided by the Brazilian National Council for Scientific and Technological Development (CNPq) for through PDJ #163055/2014-9, PVE #400325/2014-4 and Universal project #448966/2014-0. The funders had no role in study design, data collection and analysis, decision to publish, or preparation of the manuscript.

==============================
Understanding public perceptions of biodiversity is essential to ensure continued support for conservation efforts. Despite this, insights remain scarce at broader spatial scales, mostly due to a lack of adequate methods for their assessment. The emergence of new technologies with global reach and high levels of participation provide exciting new opportunities to study the public visibility of biodiversity and the factors that drive it. Here, we use a measure of internet saliency to assess the national and international visibility of species within four taxa of Brazilian birds (toucans, hummingbirds, parrots and woodpeckers), and evaluate how much of this visibility can be explained by factors associated with familiarity, aesthetic appeal and conservation interest. Our results strongly indicate that familiarity (human population within the range of a species) is the most important factor driving internet saliency within Brazil, while aesthetic appeal (body size) best explains variation in international saliency. Endemism and conservation status of a species had small, but often negative, effects on either metric of internet saliency. While further studies are needed to evaluate the relationship between internet content and the cultural visibility of different species, our results strongly indicate that internet saliency can be considered as a broad proxy of cultural interest.

Introduction

Species assessments are a central component of applied conservation science. In particular the categorisation and quantification of species richness, endemism and extinction risk has shaped modern conservation institutions and the geographies of conservation action (Ladle & Whittaker, 2011). However, far less attention has been given to measuring and understanding the cultural visibility and profile of wild animals and plants. This may be explained by a combination of the influential natural science critique of conservation strategies based on popular or iconic species (e.g. Andelman & Fagan, 2000; Simberloff, 1998), efforts to create standardised global biodiversity data based on taxon and habitat units (Bowker, 2000), and data constraints that until recently precluded systematic assessments of species ‘culturalness’ at larger geographic scales (Jepson & Ladle, 2009).

More recently however, two emerging trends are creating an imperative to generate measures of species ‘culturalness’. One is the rise of functionalism as an object of analysis in contemporary biodiversity and conservation science. This perspective views species as assemblies of traits and seeks to understand the geographies of trait distribution across space and scale (Violle et al., 2014), the role of traits in ecosystem assembly and function and the implications of their loss (Cadotte, Carscadden & Mirotchnick, 2011). Secondly, the rise of natural capital and ecosystem services policy frames that are pushing conservationists and academics to restate the value of biodiversity conservation in the quantitative language of economics. Cultural services are a component of these frames and this is creating the imperative to develop metrics that capture cultural services and quality-of-life benefits (Chan et al., 2012; Dallimer et al., 2012; Daniel et al., 2012; Helm, 2015; IPBES, 2014; Norgaard, 2010).

The cultural profile, or public popularity, of a given species derives from an interaction between its phenotypic (physical appearance, size, behaviour, etc.) and biogeographic traits, and the attitudes, values and cultural framings of the publics with which it interacts (Ducarme, Luque & Courchamp, 2013; Jepson & Barua, 2015; Lorimer, 2007). It is simultaneously a trait in itself and a proxy of the benefits that arise from the interactions of people and culture with species and nature. Until recently, measuring the relative public popularity of a species required extensive and resource intensive social surveys, significantly constraining systematic assessments of a wide range of species at broad geographic scales (e.g. Jepson & Ladle, 2009). The internet, with its global reach and high levels of social participation, has provided new opportunities for measuring public perception, visibility and interest in the environment in general (McCallum & Bury, 2013; Proulx, Massicotte & Pepino, 2014; Richards, 2013) and species in particular (Kim et al., 2014; Roberge, 2014; Żmihorski et al., 2013). Such approaches have been made possible by the vast amounts of data generated directly or indirectly though people’s interaction with the internet and the concomitant development of big data analytics (Kitchin, 2014) and offer new opportunities to generate metrics of biodiversity that are meaningful to politicians and publics who influence decision makers (Nemesio, Seixas & Vasconcelos, 2013; Żmihorski et al., 2013). For example, web proxies of public interest in a species include the number of times that a species name has been used as a search term (Kim et al., 2014; Schuetz et al., 2015) or the number of web sites that mention the name of a species (Żmihorski et al., 2013). Such approaches fall into the emerging sub-discipline of culturomics, the analysis of culture through the analysis of changes in word frequencies in large bodies of texts (Michel et al., 2011).

A body of recent research has identified a set of ecological and social factors that explain the public popularity of a species, although their relative importance in different cultural settings is largely unknown. These factors tend to fall into two main groupings: familiarity and aesthetic appeal. For example, a recent study demonstrated that internet searches for 68 resident bird species in the United States were positively associated with estimates of bird population densities, i.e. people were more interested in the birds that were familiar to them (Schuetz et al., 2015). Independently of whether a species is familiar, people also have strong biases towards larger (Knegtering, van der Windt & Schoot Uiterkamp, 2011; Ward et al., 1998; Żmihorski et al., 2013), more colourful (Lišková & Frynta, 2013; Lišková, Landová & Frynta, 2014), cuter (Borgi et al., 2014) and more human-like (Batt, 2009) species. Finally, the perceived conservation status of a species may also influence its public visibility (Clucas, McHugh & Caro, 2008).

Here, we use a metric of internet salience to assess public visibility of Brazilian bird species belonging to four taxa (hummingbirds, toucans, parrots and woodpeckers) within Brazilian and international webpages. We then test the relative importance of factors related to familiarity (range size, human population within range, occurrence in anthropogenic environments), aesthetic appeal (body size) and conservation interest (endemism, endangerment) in explaining public visibility.

Methods

We evaluated the internet saliency of 236 bird species officially occurring in Brazil (Comitê Brasileiro de Registros Ornitológicos, 2008) and belonging to four distinct groups: hummingbirds (Family Trochilidae, n = 80), toucans (Family Ramphastidae, n = 18), parrots (Family Psittacidae, n = 85) and woodpeckers (Family Picidae, n = 51). These four groups were chosen because they are highly visible, possess substantial within-family variability in size and other phenotypic characteristics, have high species richness in Brazil and all contain species that are abundant in anthropogenic landscapes.

Internet saliency of each species was assessed by performing a web search of individual species names using Google’s Custom Search API. In order to assess the Brazilian and international saliency of each species, we carried out two types of searches: one with Brazilian (Portuguese-language) popular names for webpages hosted in Brazil and one with English-language popular names for international websites. Furthermore, we restricted both searches to webpages that also mentioned the term “bird” (or “ave” for Brazilian searches) in order to reduce potential biases in the cases where the species name is also commonly mentioned in non-biological contexts (e.g. “toucans” have a political connotation in Brazil). The number of webpages returned by the search was log-transformed and ultimately used as a metric of internet saliency (Sitas, Baillie & Isaac, 2009; Żmihorski et al., 2013).

Additionally, we collected information related to public familiarity with the species (range size, human population within species range, occurrence in anthropogenic environments), aesthetic appeal (body size) and conservation interest (endemism, endangerment). Range size (RAN) was calculated as the extent of the species distribution based (km2) on BirdLife’s species distribution maps (BirdLife International and NatureServe, 2014). Human population within the species range (POP) was estimated from a gridded map of world population (Center for International Earth Science Information Network and Centro Internacional de Agricultura Tropical, 2005) by summing the values of all the map cells that intersect the species distribution. Data on species occurrence in anthropogenic environments (ANT) and body size (SIZ) was collected from available bird guides (Sigrist, 2014). Endemic species were identified from the list of Brazilian birds published by the Comitê Brasileiro de Registros Ornitológicos (2008) and endangered species were classified as all the species with an endangerment category of Vulnerable (VU), Endangered (EN) or Critically Endangered (CR) according to the IUCN Red List of Threatened Species (IUCN, 2014). The full data used for analysis is available in S1 Appendix. All explanatory variables were standardized prior to analysis (Schielzeth, 2010) and we found no evidence of severe collinearity between variables (Spearman’s r ≤ |0.75|).

The relationship between familiarity, aesthetic and conservation interest variables and internet saliency was assessed using Generalized Linear Models (GLMs) with Gaussian distribution and identity-link function. We implemented models independently for Brazilian and international internet saliency metrics and for each individual bird group as well as for all species pooled together. All possible model combinations (without interactions) relating internet saliency to the six explanatory variables were calculated using a multimodel inference approach (Burnham & Anderson, 1998) implemented with the MuMIn package for R Software. Next, we identified the best performing models according to Akaike’s Information Criterion corrected for sample size (AICc) and Akaike’s weights (ωAICc). However, because no single model clearly outperformed the others (ωAICc < 0.9 for all models and groups evaluated), we used a model averaging approach to obtain averaged parameter estimates and the relative importance of each explanatory variable. For this process, we considered only models with ωAICc ≥ 0.05 as this score can be interpreted as the probability that a given model is the best fit for the observed data, given the candidate set of models (Burnham, Anderson & Huyvaert, 2011). All the analysis were implemented in R Software v3.1.3 (R Core Team, 2015) and figures were elaborated using the ggplot2 library available for the same software package.

Results

All the species evaluated in this work had at least one webpage mention in international websites and only eight species (3% of all species) did not show any webpage mention in Brazilian websites. The average number of webpage mentions in Brazilian webpages was highest for woodpeckers (∼72 per species) and lowest for hummingbirds (∼32 per species), whereas the most and least mentioned groups in international websites were respectively parrots (∼1872 web mentions per species) and hummingbirds (∼643 web mentions per species). Internet saliency was significantly higher for international searches than for Brazilian searches (ANOVA, F = 976.8, p < 0.001), but no significant difference (ANOVA, F = 1.7, p = 0.16) was found between the different bird groups in either setting (Fig. 1).

Figure 1 Distribution of the log-transformed number of Brazilian and international webpages mentioning species in each of the studied bird taxon.

Horizontal lines indicate median values, upper and lower box hinges represent first and third quartiles, whiskers extend to 1.5 times the inter-quartile range, and dots represent values outside this range.

At the species level, the Cream-colored Woodpecker (Celeus flavus) had the highest internet saliency score in Brazilian searches whereas for international searches the highest score was obtained by the Scarlet Macaw (Ara macao). Only the Tocu Toucan (Ramphastos toco) had the highest saliency for both Brazilian and international searches within its group; the most salient species differed between searches for all the other bird groups (Table 1).

Table 1 The five most represented species for each study taxon (hummingbirds, parrots, toucans, woodpeckers) on Brazilian and international webpages.

Bird group	Rank	Brazilian webpages	International webpages	
Hummingbirds	1	Swallow-tailed hummingbird (Eupetomena macroura)	White-necked jacobin (Florisuga mellivora)	
2	Racket-tailed coquette (Discosura longicaudus)	White-throated hummingbird (Leucochloris albicollis)	
3	Gilded sapphire (Hylocharis chrysura)	Tufted coquette (Lophornis ornatus)	
4	White-throated hummingbird (Leucochloris albicollis)	Black-throated mango (Anthracothorax nigricollis)	
5	Black jacobin (Florisuga fusca)	Swallow-tailed hummingbird (Eupetomena macroura)	
Toucans	1	Toco toucan (Ramphastos toco)	Toco toucan (Ramphastos toco)	
2	Channel-billed toucan (Ramphastos vitellinus)	Channel-billed toucan (Ramphastos vitellinus)	
3	Green-billed toucan (Ramphastos dicolorus)	Chestnut-eared aracari (Pteroglossus castanotis)	
4	Saffron toucanet (Pteroglossus bailloni)	Saffron toucanet (Pteroglossus bailloni)	
5	Chestnut-eared aracari (Pteroglossus castanotis)	Green aracari (Pteroglossus viridis)	
Parrots	1	Golden parakeet (Guaruba guarouba)	Scarlet macaw (Ara macao)	
2	Blue-and-yellow macaw (Ara ararauna)	Hyacinth macaw (Anodorhynchus hyacinthinus)	
3	Blue-winged parrotlet (Forpus xanthopterygius)	Blue-and-yellow macaw (Ara ararauna)	
4	Blue-fronted amazon (Amazona aestiva)	Monk parakeet (Myiopsitta monachus)	
5	Red-spectacled amazon (Amazona pretrei)	Spix’s macaw (Cyanopsitta spixii)	
Woodpeckers	1	Cream-colored woodpecker (Celeus flavus)	Lineated woodpecker (Dryocopus lineatus)	
2	Campo flicker (Colaptes campestres)	White woodpecker (Melanerpes candidus)	
3	Lineated woodpecker (Dryocopus lineatus)	Golden-olive woodpecker (Colaptes rubiginosus)	
4	Blond-creasted woodpecker (Celeus flavescens)	Crimson-crested woodpecker (Campephilus melanoleucos)	
5	Green-barred woodpecker (Colaptes melanochloros)	Campo flicker (Colaptes campestres)	

The analysis of AICc scores and Akaike weights revealed that intercept-only models were implausible when compared with the best models (ΔAICc ≥ 14), indicating some of the predictors analysed clearly contributed to explain the variability in internet saliency between species. However, no single model is a clear best fit for the data; rather, there are several competing models with good explanatory power of both Brazilian and international internet saliency (Tables 2 and 3, respectively).

Table 2 Ranked set of best candidate models of Brazilian internet salience.

Group	Rank	Explanatory variables	AICc	ΔAICc	ωAICc	
RAN	POP	ANT	SIZ	EDM	EDG	
Hummingbirds	1		X	X		X		73.83	0	0.107	
2		X	X				73.93	0.10	0.102	
3	X	X	X				74.17	0.34	0.090	
4	X		X	X	X		74.25	0.42	0.087	
5		X	X	X			74.99	1.16	0.060	
6	X	X	X		X		75.13	1.31	0.056	
7	X	X	X	X			75.25	1.43	0.052	
Toucans	1		X		X			9.74	0	0.294	
2		X	X	X			10.89	1.16	0.165	
3		X		X	X		11.71	1.97	0.110	
4		X	X				12.14	2.40	0.088	
5	X	X		X			13.01	3.28	0.057	
6		X		X		X	13.18	3.44	0.053	
Parrots	1		X		X	X		144.25	0	0.315	
2		X	X	X	X		145.90	1.65	0.138	
3		X		X	X	X	146.11	1.85	0.125	
4	X	X		X	X		146.33	2.08	0.111	
5		X	X	X	X	X	147.65	3.40	0.058	
Woodpeckers	1		X	X	X			67.61	0	0.213	
2		X	X	X	X		69.16	1.55	0.098	
3		X	X				69.28	1.66	0.093	
4	X	X	X				69.34	1.72	0.090	
5	X	X	X	X			69.38	1.77	0.088	
6		X	X	X		X	70.15	2.54	0.060	
7		X	X		X		70.22	2.60	0.058	
All groups	1		X	X	X	X		310.30	0	0.382	
2		X	X	X	X	X	310.97	0.67	0.274	
3	X	X	X	X	X		312.08	1.78	0.157	
4	X	X	X	X	X	X	312.71	2.41	0.115	
Note:

Models are ranked by order of lowest AICc score and only models with a weight over 0.05 were considered. Rank, explanatory variables, AICc score, delta AICc relative to the model with lowest AICc score and Akaike weights are given for each individual model.

Table 3 Ranked set of best candidate models of international internet salience.

Group	Rank	Explanatory variables	AICc	ΔAICc	ωAICc	
RAN	POP	ANT	SIZ	EDM	EDG	
Hummingbirds	1	X	X					1.04	0	0.080	
2	X						1.12	0.07	0.077	
3	X	X			X		1.28	0.24	0.071	
4	X	X		X			1.96	0.91	0.051	
Toucans	1				X			15.58	0	0.206	
2		X		X			15.99	0.41	0.168	
3				X		X	16.73	1.14	0.116	
4		X		X		X	17.96	2.38	0.063	
5			X	X			18.08	2.50	0.059	
Parrots	1	X		X	X	X		127.39	0	0.123	
2			X	X	X		127.55	0.16	0.114	
3	X		X	X			127.94	0.56	0.093	
4	X			X	X		128.02	0.63	0.090	
5		X	X	X	X		128.88	1.49	0.058	
Woodpeckers	1	X	X	X	X	X		−30.50	0	0.157	
2	X		X	X	X		−30.24	0.27	0.137	
3		X	X	X	X		−29.63	0.87	0.101	
4	X	X		X	X		−29.47	1.03	0.094	
5	X			X	X		−28.43	2.07	0.056	
All groups	1	X		X	X	X		190.17	0	0.373	
2	X	X	X	X	X		191.41	1.24	0.201	
3	X		X	X	X	X	192.29	2.11	0.130	
4	X	X	X	X	X	X	193.53	3.35	0.070	
Note:

Models are ranked by order of lowest AICc score and only models with a weight over 0.05 were considered. Rank, explanatory variables, AICc score, delta AICc relative to the model with lowest AICc score and Akaike weights are given for each individual model.

Model averaging indicated that human population within the species range, presence in anthropogenic habitats, body size and endemism were important predictors of overall bird internet saliency in Brazil (Relative importance = 1.00). All these predictors related positively with internet saliency but human population within the species range showed the largest effect size for all bird groups (Table 4). In contrast, the importance of body size and presence in anthropogenic differed greatly between bird groups; presence in anthropogenic habitats had a greater effect size on hummingbirds and woodpeckers while the effect of body size was more important for toucans and parrots. The remaining predictors, endangerment status (Relative importance = 0.42) and range size (Relative importance = 0.29), were also included in some models and had a positive but negligible effect on internet saliency.

Table 4 Summary outputs of Brazilian and international internet saliency model averages.

Group	Predictor	Brazil internet saliency	International internet saliency	
Estimate	SE	ωAICc	Estimate	SE	ωAICc	
Hummingbirds	Intercept	0.98	0.06	–	2.72	0.03	–	
RAN	−0.03	0.05	0.36	0.10	0.04	1.00	
POP	0.43	0.06	1.00	0.04	0.04	0.72	
ANT	0.25	0.11	1.00	–	–	–	
SIZ	0.02	0.04	0.36	0.01	0.02	0.18	
EDM	0.08	0.12	0.45	−0.03	0.06	0.25	
EDG	–	–	–	–	–	–	
Toucans	Intercept	1.02	0.10	–	2.79	0.08	–	
RAN	<0.01	0.03	0.07	–	–	–	
POP	0.46	0.07	1.00	0.05	0.08	0.38	
ANT	0.11	0.20	0.33	0.02	0.08	0.10	
SIZ	0.20	0.11	0.88	0.32	0.09	1.00	
EDM	−0.04	0.15	0.14	–	–	–	
EDG	<0.01	0.05	0.07	−0.08	0.17	0.29	
Parrots	Intercept	1.08	0.08	–	2.87	0.09	–	
RAN	0.01	0.04	0.15	0.07	0.07	0.58	
POP	0.42	0.07	1.00	<0.01	0.03	0.47	
ANT	0.03	0.09	0.26	0.20	0.15	0.56	
SIZ	0.37	0.06	1.00	0.26	0.06	1.00	
EDM	0.34	0.14	1.00	−0.20	0.15	0.60	
EDG	0.03	0.09	0.24	–	–	–	
Woodpeckers	Intercept	0.97	0.09	–	2.73	0.04	–	
RAN	0.03	0.06	0.25	0.06	0.04	0.81	
POP	0.37	0.08	1.00	0.04	0.04	0.65	
ANT	0.36	0.14	1.00	0.07	0.06	0.73	
SIZ	0.08	0.08	0.66	0.06	0.03	1.00	
EDM	−0.04	0.12	0.22	−0.19	0.04	1.00	
EDG	<−0.01	0.07	0.09	–	–	–	
All groups	Intercept	1.00	0.04	–	2.76	0.03	–	
RAN	0.01	0.03	0.29	0.09	0.03	1.00	
POP	0.41	0.03	1.00	0.01	0.02	0.35	
ANT	0.24	0.07	1.00	0.15	0.06	1.00	
SIZ	0.22	0.03	1.00	0.17	0.02	1.00	
EDM	0.23	0.08	1.00	−0.16	0.06	1.00	
EDG	0.05	0.09	0.42	<−0.01	0.04	0.26	
Note:

Average parameter estimates and Akaike’s weights (ωAICc) of Brazilian and international internet saliency models. Model averaging was carried out including only models with Akaike’s weight scores over 0.05.

For international saliency, body size, endemism, presence in anthropogenic habitats and range size were all important predictors of overall international internet saliency (Relative importance = 1.00). While body size showed the highest effect when all species were considered together in the models, it was only marginally higher than that of endemism and presence in anthropogenic habitats (Table 4). Also, the importance of these variables changed when bird groups were analysed individually; body size was particularly important for toucans, parrots and woodpeckers, range size was important for hummingbirds and endemism was important for woodpeckers. Again, most of these predictors related positively with internet saliency with the clear exception of endemism, which showed a consistent negative relationship. Human population within the species range (Relative importance = 0.35) and endangerment status (Relative importance = 0.26) had very little importance overall internet saliency and, when included in the top models, they generally showed a negligible effect.

Discussion

The internet salience of different bird species varied widely, with a few species being characterized by very high saliency (in Brazilian or international webpages) and the majority of species having low saliency. Such a log-normal distribution is unsurprising and probably reflects the limited number of species that have a public profile that goes beyond their conservation or ecological status. A more detailed analysis of the most salient species suggests that these birds are often kept as pets or have, for some reason, become part of popular culture (nationally or internationally). For example, in Brazilian web-sites, two of the most salient parrot species (Guaruba guarouba, Amazona pretrei) are highly sought after cage birds in Brazil (Nobrega Alves, De Farias Lima & Araujo, 2013). Another example is the Cream-colored Woodpecker (Celeus flavus), which was the most salient woodpecker in Brazilian web-sites. This species gives name to a classic children’s book series (Sítio do Picapau Amarelo) written by Monteiro Lobato that was later adapted for theatre and television. In contrast, the three most salient parrots on international websites (Ara macao, Anodorhynchus hyacinthinus, Ara ararauna) are very large, impressive and colourful macaws commonly exhibited in zoos internationally and with considerable ecotourism appeal. There is long history of procurement and fascination of western bird collectors with such colourful species (Boehrer, 2010; Watson et al., 2015), and our results suggest that human interest in these species is still very high. It is also interesting to note that the ecologically extinct Spix’s Macaw (Cyanopsitta spixii) has a higher saliency in international web sites, possibly because of its status as a global conservation icon (Juniper, 2004) and as inspiration for the central characters in the recent animated movies Rio and Rio 2 (Yong, Fam & Lum, 2011).

As might be anticipated, factors associated with familiarity (as measured by potential for personal encounters) were associated with higher saliency in Brazilian websites. Specifically, the human population within the range of a species seems to be the most important variable explaining internet saliency, although species presence in anthropogenic habitats was also an important predictor. This is true for each group individually and for all four taxa when analysed together. Our findings support the importance of local familiarity in determining popularity or awareness of bird species and are also concordant with studies of internet search behaviour (Schuetz et al., 2015). Such concordance of findings also suggests that internet content production and search behaviour may be driven by similar socio-cultural factors.

In contrast, the main driver of the overall internet saliency of Brazilian bird species in international websites was body size. Although the importance of body size was lower when bird groups comprised of relatively small species with little variation in size (e.g. Hummingbirds and Woodpeckers) were considered individually, its importance was also evident when all bird species were evaluated together. This strongly suggests that in the absence of direct experience, preferences or awareness of particular species is influenced by phenotypic characteristics. Body size has previously been demonstrated to influence internet saliency of bird species (Żmihorski et al., 2013) and the attractiveness of zoo animals (Frynta et al., 2013). Of course, other phenotypic characteristics (e.g. colourfulness, attractiveness of song, behaviours, etc.) may be even more important, but are much harder to assess in the absence of in depth social surveys. It should also be mentioned that size was also significantly associated with internet saliency of bird species in Brazilian webpages, although in this case it is superseded by familiarity. Interestingly, presence in anthropogenic habitats was also an important predictor of international bird saliency, particularly for bird groups with smaller body sizes (hummingbirds and woodpeckers). While a link with familiarity, in a broader sense, is unlikely to exist, this result suggests that opportunistic encounters between bird species and international visitors to Brazil may also contribute to international internet saliency.

From a conservation perspective, the results for endemism are particularly interesting and may have important implications for policy. The fact that Brazilian endemics are associated with higher internet saliency strongly suggests that this characteristic provides a higher profile for these species within the national setting. However, the negative association between international internet saliency and endemism implies that these perceptions do not extend beyond national boundaries. Raising awareness of these species within the international community may bring conservation benefits, given that studies have shown that international tourists are more willing to financially support the conservation of endemic species (Veríssimo et al., 2009).

It is also notable that conservation status did not significantly influence internet saliency of bird species in Brazilian or international webpages. This is probably less a reflection of a failure of the conservation movement to publicise species at risk, and more an indication of the over-riding importance of familiarity (for Brazilian nationals) and phenotypic characteristics (for the international community). The exceptions to these general trends are species that have become global icons, such as Spix’s Macaw, or which are both endangered and phenotypically appealing such as the Hyacinth Macaw.

Overall, our findings suggest that species ‘culturalness,’ here represented by internet salience, is a relational trait that emerges from a package of species traits that afford humans something. Based on this study, key species traits seem to be behavioural, dietary and/or phenotypic traits that afford a) taming and easy husbandry, b) close and/or regular viewing of the species during the everyday lives of ‘general’ publics, c) a sense of awe, wonder and/or ‘exoticism,’ and d) the creation of animal characters in stories. Nevertheless, it should be noted that other taxa may possess or exhibit traits that generate negative perceptions that may contribute to their visibility, such as ferocious behaviour, venomous or poisonous attributes and invasive characteristics. From a conservation perspective, the sentiment associated with a species (positive or negative) is probably as relevant as its cultural salience (high or low visibility within a cultural context). For example, Xu and colleagues showed that perceptions towards wolves tend to be negative in Tibet, despite generally positive views about nature conservation in Tibetan society (Xu, Yang & Dou, 2015). Even when a species has cultural visibility and is positively perceived, other deeply ingrained cultural practices such as hunting may undermine its conservation (Gama et al., 2016). These examples clearly illustrate some of the implications that cultural perceptions can have on the success of conservation policies and actions.

Ultimately, there is a need for conservationists to assess the multiple dimensions of species interactions with culture. Although the culturomic techniques presented here only addresses a single dimension, our work demonstrates the potential of new digital techniques for investigating these complex interactions. Future research in these areas will benefit from an increased collaboration and engagement with social scientists (Ehrlich, 2002), the digital humanities and computer scientists working in the emerging fields of natural language processing and text sentiment analysis (Wagner-Pacifici, Mohr & Breiger, 2015).

Conclusions

Assessing and understanding internet saliency of biodiversity components is important because it can be thought of as a broad proxy of cultural visibility, incorporating elements of cultural identity, heritage, spiritual significance, inspiration and aesthetic appreciation, recreation, and tourism (Dallimer et al., 2014). These elements are harder to systematically and directly quantify than more tangible biodiversity values (e.g. economic value), but play a key role in conservation efforts (Prokop & Fančovičová, 2013; Żmihorski et al., 2013). This is because, in general, people are more likely to support conservation of a species (and other components of the natural world) with characteristics that they value (Prokop & Fančovičová, 2013).

This study represents one of the first attempts to capture and understand the cultural value of species using internet saliency and a ‘big data’ approach. With the global expansion of digital culture (Gere, 2008), there is immense potential to expand the application of digital tools to conservation science, for example to support conservation prioritisation and planning, valuation of cultural ecosystem services, and the development of trait-based ecology. In contrast to ‘traditional’ biodiversity sets that take long-time periods to compile, model and apply, culturomic approaches once developed can be semi-automated enabling assessments of at finer temporal and spatial resolutions. This would potentially enable conservation science to more effective respond to public opinion and thereby strengthen its democratic legitimacy. However, more studies are needed to convincingly demonstrate the connections between internet content and the behaviour of internet users and cultural values.

Supplemental Information

Supplemental Information 1 S1 Appendix. Dataset of species level variables used for analysis.

Dataset includes Brazilian and international internet salience scores, range size, human population within the range, presence in anthropogenic habitats, body size and endemic or endangered status for each individual species considered for analysis.

Click here for additional data file.

Additional Information and Declarations

Competing Interests

Author Contributions

Data Deposition

The authors declare that they have no competing interests.

Ricardo A. Correia conceived and designed the experiments, performed the experiments, analyzed the data, wrote the paper, prepared figures and/or tables.

Paul R. Jepson conceived and designed the experiments, wrote the paper.

Ana C. M. Malhado conceived and designed the experiments, wrote the paper.

Richard J. Ladle conceived and designed the experiments, wrote the paper.

The following information was supplied regarding data availability:

All the data used for the paper is made available in the Supplementary Material.

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
