# Peer review of "Familiarity breeds content: assessing bird species popularity with culturomics"

_PeerJ, doi:10.7717/peerj.1728_

## Round 0.1 · original submission · Minor Revisions

The reviewer's and I agree that a better explanation of the difference between how "popular" a species is on the internet and its cultural biodiversity value is needed. For example some species are popular because they are invasive. A bit more on this and how you might separate cultural value from mere popularity would make the piece better and acceptable for publication. See in particular reviewer #2's comments.

Reviewer 1 ·

Basic reporting

Text is clear and well-written

Experimental design

Good design for assessing "salience," but see comments on Validity below.

Validity of the findings

Abstract:
“Understanding public perceptions of biodiversity is essential to ensure continued support for conservation efforts.....While further studies are needed to evaluate the relationship between internet content and sentiment towards different species, our results strongly indicate that internet saliency can be considered as a broad proxy of cultural interest.” (my underline)

It would be helpful if the text/abstract made it more clear that this study assesses “saliency” (interest, prominence, etc.) and not “sentiment” which is critical for “support of conservation efforts” ? For exampIe, I expect one could find a high count of webpage references to “Salt Cedar (Tamarisk),” an invasive species which has adverse impacts on biodiversity in the American Southwest.” This is high “public visibility”, but it is certainly an “unpopular” species.

I am not so sure how salience/interest relate to “perceptions of biodiversity.” It might be helpful if the authors could be more thorough, thoughtful and explicit on how the understanding of factors behind salience can be used to create the need and programs to support conservation and biodiversity. (I say this in a constructive way, and not to reflect on the quality of the study.)

Conclusions:
“Assessing and understanding internet saliency of biodiversity components is important because it can be thought of as a broad proxy of cultural value...” Related to the above comments, I do not see how “salience” can be a proxy for “value” unless the webpage references are designated as considering a species as “good” or “bad.”

Additional comments

This is a very interesting and potentially useful avenue of research. But I would like to see more on how salience relates to biodiversity policies, especially since salience can refer to culturally considered "good" and "bad" species

·

Basic reporting

No comments

Experimental design

No comments

Validity of the findings

This article is well written, well structured, and uses solid analytical methods to consider an interesting issue. But I struggle with the interpretation of results – summarized in the conclusion as indicating that it might be possible to use internet saliency as an indicator of the cultural value of species of conservation concern.

In fact, I worry about the idea of using internet saliency in that way.

Why? If I am understanding correctly, the results suggest that it is NOT conservation status which influences internet saliency, rather it is ‘familiarity’ (most notably indicated by presence of animal in human environment, also associated with species range, population) - although body size also important (endemism also important in Brazil). In layman’s language, this indicates that the most ‘popular’ species (most frequently referred to on the internet) are the ones people see most frequently (large, and nearby). Within country, the most popular species are also those which are unique/special to the local environment – I’ll come back to that point later.
Large and frequently ‘seen’ species would presumably also include species such as cattle and sheep which are nurtured primarily for consumptive purposes in addition to species kept as household pets (including dogs and cats). Species that are likely to have the MOST internet saliency are thus, potentially, common species which are not in need of protection. Valuation is all about relativities (which species are most / least ‘valuable). Rather than offering protection to ‘rare’ species, if one were to use internet saliency as a proxy for species ‘value’ one would run the risk of further endangering some species – particularly those that are not large, and not in frequent contact with humans.
I thus see this article, not as one which has demonstrated an alternative method to ‘support conservation prioritization and planning, valuation of ecosystem services’ but rather as one that has demonstrated the absolute need for biophysical scientists to work with social marketers or similar, to help increase public awareness of the importance of species (and ecosystems) that are neither large nor common and which are thus not perceived by the public as being ‘valuable’. This is likely challenging, given the potential for ‘overload’ (people throughout the world are likely unable to have the time, money or emotional capacity to worry about ALL species; some means of prioritizing needs to be found)…..
Which brings me to the other interesting point highlighted in this article, namely that endemism is important within country. Perhaps social marketers or similar could also help us to understand ways to capitalize on that observation, better harnessing feelings of national identity and patriotism towards species protection (rather than towards hatred of people of other nations or of other religions). That could get around the problem of ‘overload’, but still find ways of protecting endemic species (even those that are not large and do not always interact with humans).

Additional comments

I’d like to see a more sophisticated discussion of the implications of results.The current discussion focuses on cultural values and valuation, but doesn’t connect (deeply) to the literature on cultural values or on valuation. Neither does it make a deep connection to other branches of literature likely to be relevant here – namely social psychology and social marketing (both concerned with what makes things ‘popular’). This 'missing' connection could end up with unintended consequences (e.g. internet saliency being used to 'prove' that endangered species are relatively unimportant). Adding that connection, could yield some interesting insights.

Reviewer 3 ·

Basic reporting

No major comments.

A re-examination of the paragraph from lines 52-60: the last sentence appears to be a fragment.

Experimental design

Knowledge gap identified, and authors state how study fills that gap. Could use a sentence explaining why Brazilian species were chosen, instead of species from a different nation.
Research question defined but no hypothesis given. It's absence does not detract from the paper, however.
Research methods clearly described.

Validity of the findings

Data included.
Discussion is very well explained, drawing upon the results and the literature to provide insight.

Additional comments

The discussion I found to be particularly well done, bringing in the literature to support the results of the study to provide a well-founded explanation of the possible drivers of "cultural value" for these Brazilian bird species.

---

## Round 0.2 · accepted · Accept

You have responded to the reviewer's suggestions adequately.